

# An algorithm for U–Pb geochronology by secondary ion mass spectrometry

Pieter Vermeesch[1]

[1]London Geochronology Centre, Department of Earth Sciences, University College London, London WC1E 6BT, United Kingdom

**Correspondence:** Pieter Vermeesch (p.vermeesch@ucl.ac.uk)

**Abstract.** Secondary Ion Mass Spectrometry (SIMS) is a widely used technique for in-situ U–Pb geochronology of accessory minerals. Existing algorithms for SIMS data reduction and error propagation make a number of simplifying assumptions that degrade the precision and accuracy of the resulting U–Pb dates. This paper uses an entirely new approach to SIMS data processing that introduces the following improvements over previous algorithms. First, it treats SIMS measurements as compositional data, using logratio statistics. This means that, unlike existing algorithms, (a) its isotopic ratio estimates are guaranteed to be strictly positive numbers, (b) identical results are obtained regardless of whether data are processed as normal ratios (e.g. $^{206}Pb/^{238}U$) or reciprocal ratios (e.g. $^{238}U/^{206}Pb$), and (c) its uncertainty estimates account for the positive skewness of measured isotopic ratio distributions. Second, the new algorithm accounts for the Poissonian noise that characterises Secondary Electron Multipliers (SEM). By fitting the SEM signals using the method of maximum likelihood, it naturally handes low intensity ion beams, in which zero counts signals are common. Third, the new algorithm casts the data reduction process in a matrix format, and thereby captures all sources of systematic uncertainty. These include significant inter-sample error correlations that arise from the commonly used Pb/U–UO$_{(2)}$/U calibration curve. The new algorithm has been implemented in a new software package called `simplex`. `simplex` was written in `R` and can be used either online, offline or from the command line. The program can handle SIMS data from both Cameca and SHRIMP instruments.

## 1 Introduction

Secondary Ion Mass Spectrometry (SIMS) combines high sensitivity with high mass resolution (Williams, 1998). This allows the technique to obtain precise U–Pb dates on ng-sized samples, whilst resolving isobaric interferences on $^{204}Pb$ to a degree that is currently unachievable by other techniques such as Laser Ablation Inductively Coupled Plasma Mass Spectrometry (LAICPMS). There are some other differences between LAICPMS and SIMS as well. LAICPMS instrumentation is built by numerous manufacturers. Popular data reduction codes such as `Iolite`, `Glitter` and `LADR` are compatible with all their output files. This facilitates the intercomparison of different laboratories, different instrument designs and so forth. In contrast, the SIMS U–Pb world is dominated by just two manufacturers. The data reduction protocols for SHRIMP (Sensitive High Resolution Ion Micro-Probe) and Cameca instruments are completely separate. Most SHRIMP laboratories use `Squid`





(Ludwig, 2000; Bodorkos et al., 2020), which is incompatible with Cameca data. In contrast, Cameca data tends to be processed
by in-house software such as M. J. Whitehouse's NordSIM spreadsheet, which are incompatible with SHRIMP data.

This paper introduces a unified algorithm for SIMS U–Pb data reduction that aims to address five problems with existing data reduction methods. The first three of these problems are:

1. Accuracy: existing algorithms give (slightly) different results depending on whether the raw data are processed as $^{206}$Pb/$^{238}$U-ratios or as $^{238}$U/$^{206}$Pb-ratios, say.

2. Precision: current data reduction routines produce symmetric confidence intervals, which are unrealistic for low intensity ion beams.

3. Systematic uncertainties: current data reduction protocols use a hierarchical error propagation approach, in which random uncertainties and systematic uncertainties are propagated separately. However such a clean separation is not always possible, and this can complicate higher order data processing steps such as isochron regression and averaging.

Sections 2 – 4 will provide further details about these problems, using synthetic examples. Section 5 will show that problems 1, 2 and 3 can be solved by treating the U–Pb system as a *compositional* data space, using logratio statistics. However logratio statistics does not solve the remaining two problems:

4. Blanks: background correction of low intensity signals such as $^{204}$Pb sometimes exceeds 100%, producing physically impossible negative isotope ratios.

5. Zeros: Pb-isotopes are usually measured by secondary electron multipliers (SEMs), which record ions as counts. For $^{204}$Pb and other low intensity ion species, it is not uncommon to register zero counts during any given analytical cycle. This causes problems if the zero count appears in the denominator of an isotopic ratio.

These problems are discussed in more detail in Sections 6 and 7. Section 8 addresses them by incorporating multinomial counting statistics into the compositional data framework. Section 13 applies the new data reduction paradigm to two datasets produced by Cameca and SHRIMP instruments, and Section 14 introduces a computer code called `simplex` that generated these results. Finally, further details about the implementation of the new algorithm are reported in an appendix to this paper.

## 2 Accuracy

The $^{206}$Pb/$^{238}$U-age ($t$) is given by:

$$t = \frac{1}{\lambda_{238}} \ln\left(1 + \frac{\left[\frac{^{206}\text{Pb}}{^{204}\text{Pb}}\right] - \left[\frac{^{206}\text{Pb}}{^{204}\text{Pb}}\right]_c}{\left[\frac{^{238}\text{U}}{^{204}\text{Pb}}\right]}\right) \tag{1}$$

where the subscript $c$ marks the common lead composition, and $\lambda_{238}$ is the decay constant of $^{238}$U. Equation 1 does not depend on the absolute amounts of $^{204}$Pb, $^{206}$Pb and $^{238}$U, but only on their ratios. Unfortunately, the statistical analysis of the ratios of



strictly positive numbers is full of potential pitfalls, as will be illustrated with an example that was inspired by McLean et al. (2016). Consider a simple dataset of ten synthetic U–Pb measurements:

| $^{238}$U | 215.9 | 208.9 | 212.4 | 186.3 | 217.8 | 196.7 | 216.4 | 171.8 | 216.0 | 200.1 |
|---|---|---|---|---|---|---|---|---|---|---|
| $^{206}$Pb | 18.45 | 12.40 | 21.35 | 62.22 | 21.35 | 45.08 | 26.65 | 75.88 | 29.02 | 11.40 |
| $^{204}$Pb | 0.1570 | 0.2870 | 0.1627 | 0.01425 | 0.1092 | 0.08175 | 0.06900 | 0.02250 | 0.04975 | 0.3850 |

**Table 1.** A synthetic U-Pb dataset, in arbitrary units (e.g, fmol, mV, mA or kHz).

Let us calculate the $^{206}$Pb/$^{238}$U-ratios for these data, which are needed to solve Equation 1. Comparing these ratios with their reciprocals yields two new sets of ten numbers:

| $^{206}$Pb/$^{238}$U | 0.085 | 0.059 | 0.101 | 0.334 | 0.098 | 0.229 | 0.123 | 0.442 | 0.134 | 0.057 |
|---|---|---|---|---|---|---|---|---|---|---|
| $^{238}$U/$^{206}$Pb | 11.70 | 16.85 | 9.95 | 2.99 | 10.20 | 4.36 | 8.12 | 2.26 | 7.44 | 17.55 |

The elementary rules of mathematics dictate that $1/(y/x) = x/y$ for any two numbers $x$ and $y$. In other words, the reciprocal of the reciprocal ratio equals that ratio. Indeed, for our example it is easy to see that $1/0.085 = 11.70$ and so forth. However, when we take the arithmetic means of the (reciprocal) ratios:

$$\left(\overline{^{206}\text{Pb}/^{238}\text{U}}\right)_a = \sum_{i=1}^{10} \left(^{206}\text{Pb}/^{238}\text{U}\right)_i /10 = 0.166, \text{ and}$$

$$\left(\overline{^{238}\text{U}/^{206}\text{Pb}}\right)_a = \sum_{i=1}^{10} \left(^{238}\text{U}/^{206}\text{Pb}\right)_i /10 = 9.14$$

then we find that

$$\frac{1}{\left(\overline{^{206}\text{Pb}/^{238}\text{U}}\right)_a} = \frac{1}{0.166} = 6.01 \neq 9.14 = \left(\overline{^{238}\text{U}/^{206}\text{Pb}}\right)_a$$

So the reciprocal of the mean reciprocal ratio does *not* equal the mean of that ratio! This is a counter-intuitive and clearly wrong result. Unfortunately, current algorithms for SIMS data reduction average ratios using the arithmetic mean, or perform (linear) regression through ratio data, which causes similar problems (Ogliore et al., 2011). Inaccurate $^{206}$Pb/$^{238}$U-ratios inevitably result in inaccurate U–Pb dates. Therefore, the numerical example shown in this section is deeply troubling for isotope geochemistry in general and SIMS U–Pb geochronology in particular.

## 3 Precision

Traditionally, the precision of isotopic data used in U–Pb geochronology has been calculated as symmetric confidence intervals. Unfortunately, this is fraught with similar problems as those discussed in Section 2. For example, take the arithmetic mean ($\bar{x}$) and standard deviation ($s_x$) of the $^{206}$Pb/$^{204}$Pb ratios in Table 1, and construct a studentised 95% confidence interval for $\bar{x}$:

| $x$ | $\bar{x}$ | $s_x$ | $\bar{x} + \frac{s_x}{\sqrt{10}}t_9^{2.5}$ | $\bar{x} + \frac{s_x}{\sqrt{10}}t_9^{97.5}$ |
|---|---|---|---|---|
| $^{206}$Pb/$^{204}$Pb | 978 | 1554 | -134 | 2090 |





where $t_{df}^{\alpha}$ is the $\alpha$-percentile of a t-distribution with $df$ degrees of freedom. Then the lower limit of the confidence interval is negative, which is physically impossible. This nonsensical result is yet another indication that there are some fundamental problems with the application of 'conventional' statistical operations to isotopic data. These problems cast doubt on the reliability of the analytical uncertainty assigned to U–Pb dates.

## 4 Systematic errors

The statistical uncertainty of analytical data can be classified into two components (Renne et al., 1998):

1. Random (or internal) errors are caused by electronic noise in the ion detectors, counting statistics, temporal variability of the blank as a result of changes in the lab environment, etc. They are *independent* for different aliquots of the same sample, and can be quantified by taking replicate measurements. The standard error of these measurements ($\sigma/\sqrt{N}$ where $\sigma$ is the standard deviation of $N$ replicate measurements) is a measure of their *precision*. The standard error can be reduced to arbitrarily low levels by simply averaging more measurements (i.e., by increasing $N$). For example, the precision of SIMS $^{206}$Pb/$^{204}$Pb-ratio measurements can be increased by simply extending the duration of the primary ion bombardment.

2. Systematic (or external) errors include the effects of decay constant uncertainty, the $^{206}$Pb/$^{238}$U-ratio of age standards etc. Getting these constants wrong causes *bias* in some or all of the measurements and thus affects the *accuracy* of the age determinations. As their name suggests, the systematic uncertainties are not independent but *correlated* between different aliquots and samples. They cannot be reduced by simple averaging.

Great care must be taken which sources of uncertainty should or should not be included in the error propagation. In some cases, inter-sample comparisons of SIMS U–Pb data may legitimately ignore systematic uncertainties. However, when comparing a SIMS U–Pb date with, say, a TIMS U–Pb or $^{40}$Ar/$^{39}$Ar age, both random and systematic uncertainties must be accounted for. The conventional way to tackle both types of comparisons is called 'hierarchical' error propagation (Renne et al., 1998; Min et al., 2000; Horstwood et al., 2016). Under this paradigm, the random uncertainties are processed first, and the systematic uncertainties afterwards.

Hierarchical error propagation is straightforward in principle but not always in practice. Some processing steps are of a hybrid nature, including both systematic and random uncertainties. $^{206}$Pb/$^{238}$U calibration for SIMS is a good example of this. $^{206}$Pb/$^{238}$U-ratios are sensitive to elemental fractionation in SIMS analysis (see Section 11 for further details). These fractionation effects are captured by the following power law (Williams, 1998; Jeon and Whitehouse, 2015):

$$\ln\left[\frac{^{206}\text{Pb}^{+}}{^{238}\text{U}^{+}}\right]_{m} = A + B\ln\left[\frac{^{238}\text{U}^{16}\text{O}_{(2)}^{+}}{^{238}\text{U}^{+}}\right]_{m} \tag{2}$$

where the subscript $m$ stands for the measured signal ratio, which is generally different from the atomic ratio. The atomic $^{206}$Pb/$^{238}$U-logratio of a sample is determined by (1) determining the intercept ($A$) and slope ($B$) of a standard ($st$) of known





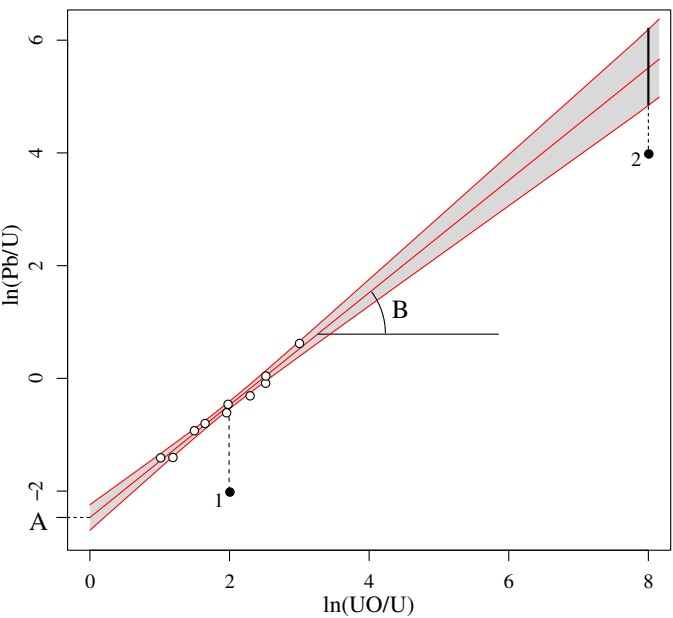

**Figure 1.** SIMS U–Pb calibration curve. White circles mark the isotopic measurements of the reference standard, black circles those of two aliquots of the same sample. The uncertainty of the linear fit is shown as a 95% confidence interval (grey area). This uncertainty can be propagated into the Pb/U-composition of the sample. It is a systematic uncertainty in the sense that it affects both aliquots. But it does not do so to the same degree. The calibration error of aliquot 2 is greater than that of aliquot 1, due to its horizontal offset relative to the calibration data.

Pb/U-ratio[1], and (2) using this calibration curve to estimate the equivalent standard Pb/U-logratio corresponding to the $UO_{(2)}$/U-logratio of the sample ($sm$). Then:

$$\ln\left[\frac{^{206}\text{Pb}}{^{238}\text{U}}\right]_a^{sm} = \ln\left[\frac{^{206}\text{Pb}}{^{238}\text{U}}\right]_a^{st} + \ln\left[\frac{^{206}\text{Pb}^+}{^{238}\text{U}^+}\right]_m^{sm} - A - B\ln\left[\frac{^{238}\text{U}^{16}\text{O}_{(2)}^+}{^{238}\text{U}^+}\right]_m^{sm} \qquad (3)$$

where the subscript $a$ stands for the estimated (for the sample) or known (for the standard) atomic logratios. The analytical uncertainty of $\ln\left[^{206}\text{Pb}/^{238}\text{U}\right]_a^{sm}$ depends on the analytical uncertainties of both the intercept ($A$) and slope ($B$) of the standard. But it does not necessarily do so to the same degree for all samples. Samples that have similar $UO_{(2)}$/U-ratios as the standard will be less affected by the uncertainty of the standard fit than samples that have very different $UO_{(2)}$/U-ratios (Figure 1). Hence it is not possible to make a clean separation between random and systematic uncertainties.

## 5  U–Pb geochronology as a compositional data problem

Section 2 pointed out that the U–Pb age equation (Equation 1) does not depend on the absolute amounts of $^{204}$Pb, $^{206}$Pb and $^{238}$U, but only on their relative abundances. Thus we could normalise the $^{204}$Pb, $^{206}$Pb and $^{238}$U measurements of Table 1 to unity and plot them on a ternary diagram. The same is true for other geochronometers such as U–Th–He and $^{40}$Ar/$^{39}$Ar (Vermeesch, 2010, 2015). In mathematics, the ternary sample space is known as a two-dimensional *simplex*. Data that live within this type of space are called *compositional* data.

---

[1]If the standard contains variable amounts of common lead, then the left hand side of Equation 2 needs to be corrected for that before applying the calibration.




Ternary systems are common in igneous petrology (e.g., the A–F–M diagram) and sedimentary petrography (e.g., the Q–F–L diagram). Geologists have long been aware of the problems associated with averages, confidence regions, and linear regression in these closed dataspaces (Chayes, 1949, 1960). But a general solution to this conundrum was not found until the 1980s, when the Scottish statistician John Aitchison published a landmark paper and book on the subject (Aitchison, 1982, 1986).

In this work, Aitchison proved that all the problems associated with the statistical analysis of compositional data can be solved by mapping those data from the simplex to a Euclidean space by means of a logratio transformation. For example, given the ternary system $\{x, y, z\}$, we can define two new variables $\{u, v\}$ so that:

$$u = \ln(x/z) \text{ and } v = \ln(y/z) \tag{4}$$

In this space, Aitchison showed, one can safely calculate averages and confidence limits. Once the statistical analysis of the transformed data has been completed, the results can then be mapped back to the simplex by means of an inverse logratio transformation:

$$x = \frac{e^u}{1 + e^u + e^v}, y = \frac{e^v}{1 + e^u + e^v} \text{ and } z = \frac{1}{1 + e^u + e^v} \tag{5}$$

For example, the $^{204,6}$Pb–$^{238}$U system of Table 1 can be mapped from the ternary diagram to a bivariate $\ln(^{204}$Pb/$^{238}$U)–$\ln(^{206}$U/$^{238}$U)-space:

| $u = \ln\left[\frac{^{206}\text{Pb}}{^{238}\text{U}}\right]$ | -2.46 | -2.82 | -2.30 | -1.10 | -2.32 | -1.47 | -2.09 | -0.82 | -2.01 | -2.87 |
| $v = \ln\left[\frac{^{204}\text{Pb}}{^{238}\text{U}}\right]$ | -7.23 | -6.59 | -7.17 | -9.48 | -7.60 | -7.79 | -8.05 | -8.94 | -8.38 | -6.25 |

Alternatively, we could also use $^{206}$Pb as the denominator isotope:

| $u = \ln\left[\frac{^{238}\text{U}}{^{206}\text{Pb}}\right]$ | 2.46 | 2.82 | 2.30 | 1.10 | 2.32 | 1.47 | 2.09 | 0.82 | 2.01 | 2.87 |
| $v = \ln\left[\frac{^{204}\text{Pb}}{^{206}\text{Pb}}\right]$ | -4.77 | -3.77 | -4.88 | -8.38 | -5.28 | -6.31 | -5.96 | -8.12 | -6.37 | -3.39 |

Calculating the average of the transformed data and mapping the results back to the simplex using the inverse logratio transformation yields the *geometric* mean of the ratios:

$$\left(\overline{^{238}\text{U}/^{206}\text{Pb}}\right)_g = 7.58 = \frac{1}{0.13} = \frac{1}{\left(\overline{^{206}\text{Pb}/^{238}\text{U}}\right)_g}$$

which is an altogether more satisfying result than in Section 2. Moving on to the 95% confidence intervals of the $^{206}$Pb/$^{204}$Pb-ratios, we first determine the conventional confidence limits for the logratios:

| $u$ | $\bar{u}$ | $s_u$ | $\bar{u} + \frac{s_u}{\sqrt{10}}t_9^{2.5}$ | $\bar{u} + \frac{s_u}{\sqrt{10}}t_9^{97.5}$ |
|---|---|---|---|---|
| $\log(^{206}\text{Pb}/^{204}\text{Pb})$ | 5.72 | 1.66 | 4.53 | 6.91 |

After the inverse-logratio transformation, these values produce an asymmetric 95% confidence interval for the geometric mean $^{206}$Pb/$^{204}$Pb-ratio of $305^{+695}_{-212}$. This interval contains only strictly positive values, solving the problem of Section 3. The




logratio trick can easily be generalised to more than three components. For example, if $^{207}$Pb is added to the mix, then the four-component $^{204|6|7}$Pb–$^{238}$U-system can be mapped to the three component $\ln\left(^{204|6|7}\text{Pb}/^{238}\text{U}\right)$-space (Figure 2).

The compositional nature of isotopic data embeds a covariant structure into very DNA of geochronology: in a $K$-component
system, increasing the absolute amount of one of the components automatically lowers the relative amount of the remaining $(K\text{-}1)$ components (Chayes, 1960). To deal with this phenomenon, it is customary in compositional data analysis to process data in matrix form, using the full covariance matrix. This approach is now widely used in sedimentary geology, geochemistry and ecology (e.g., Weltje, 2002; Vermeesch, 2006; Pawlowsky-Glahn and Buccianti, 2011), and has recently been adopted for geochronological applications as well (Vermeesch, 2010, 2015; McLean et al., 2016). The logratio covariance matrix approach
is also uniquely suited to capture the systematic uncertainties (i.e. the inter-sample error correlations) that are produced by the SIMS U–Pb calibration procedure (Section 4).

In conclusion, the logratio transformation solves the statistical woes described in Sections 2, 3 and 4 of this paper. However there are two additional problems that require further remediation.

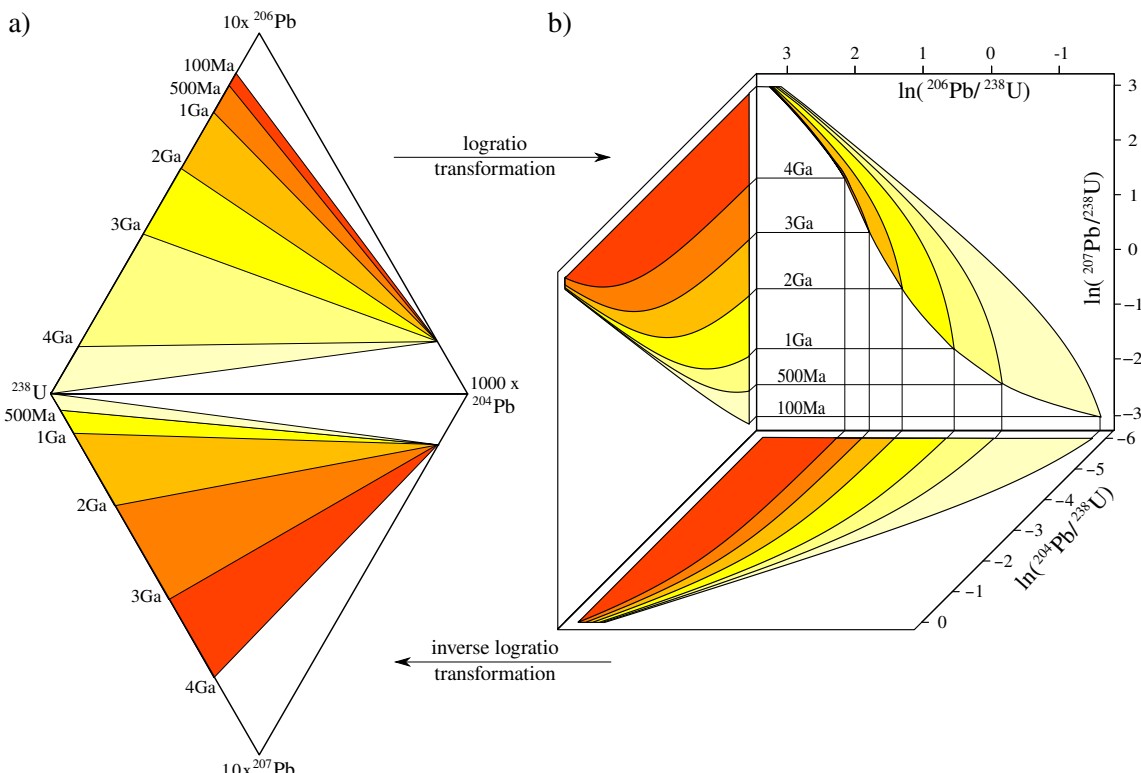

**Figure 2.** The U–Pb age equation (a) shown on the four-component simplex, and (b) mapped to a three-dimensional Euclidean logratio space. $^{235}$U is omitted from the diagrams because it exists in a constant ratio to $^{238}$U.





## 6  Blanks

Many mathematical operations are easier in logarithmic space than in linear space: multiplication becomes addition, division becomes subtraction, and exponentiation becomes multiplication. These mathematical operations are very common in mass spectrometer data processing chains (Vermeesch, 2015). However there are exceptions. For example, blank correction does not involve the multiplication but subtraction of two signals. For low intensity ion beams such as [204]Pb it is possible, by chance, that the background exceeds the signal. This results in negative values of which one cannot take the logarithm.

The subtraction problem can be solved by using a different logarithmic change of variables:

$$\beta_x^y \equiv \ln\left(\frac{y-b}{x-b}\right) \tag{6}$$

where $x$ and $y$ are the signals and $b$ is the blank. The infinite space of $\beta_x^y$ covers all possible values of $x$, $y$ and $b$ for which $x > b$ and $y > b$. Thus, blank correction should be done in $\beta$-space, given an appropriate error model as described in Section 8.

This black correction method does not account for isobaric interferences, which may result in 'overcounted' signals. The 165  high mass resolution of SIMS instruments removes most but not all isobaric interferences. For example, spurious HfSi, REE dioxide, or long-chain hydrocarbon ions can interfere with [204]Pb, which is generally the rarest isotopic species detected. If unaccounted for, these interferences lead to the proportion of non-radiogenic Pb being overestimated (and the proportion of radiogenic Pb underestimated), resulting in excessive common Pb corrections, and underestimated dates.

The accuracy of the blank measurements can monitored via the use of isotopically homogeneous reference materials (Black, 170  2005). A correction can then be applied by choosing the 'session blank' that brings the common-Pb corrected [207]Pb/[206]Pb-ratios in alignment with the reference values.

## 7  Zeros

SIMS instruments can contain both Faraday and Secondary Electron Multiplier (SEM) detectors. Faraday detectors record ion beams as electrical voltages, i.e. as decimal numbers that can either be positive or negative. In contrast, SEM signals are 175  registered as discrete counts, i.e. as integers. Unfortunately such count data are incompatible with the logratio transformation. For example, it is not uncommon for SEM detectors to register zero counts for low intensity ion beams such as the blank and [204]Pb. These zero counts blow up the logratio transformation, because log(0)=$-\infty$.

This and other issues are diagnostic of a fundamental difference between compositional data and counting data that has been previously recognised and solved in fission track dating (Galbraith, 2005) and in sedimentary point counting (Vermeesch, 180  2018b). The same solutions can be applied to mass spectrometric count data in general, and to U–Pb geochronology in particular (Section 8).




# 8   Dealing with count data

Standard data reduction procedures for geochronology assume normally distributed residuals. In compositional data analysis, these are replaced by logistic normal distributions. However, neither the normal nor the logistic normal distribution are perfectly suited for dealing with discrete count data. The multinomial distribution is a simple alternative that seems better suited for the task at hand. Before we proceed, let us define the following variables:

- $\phi_x$ and $\phi_b$: the normalised true ion beam intensities (in counts per second) of mass $x$ (from a set of monitored masses $X$) and the normalised background signal, respectively, so that

$$\phi_b + \sum_{x \in X} \phi_x = 1 \tag{7}$$

- $d_x, d_b$: the dwell times of mass $x$ and the background $b$

- $\theta_x$ and $\theta_b$: the normalised expected beam *counts* of the ions and the background, so that

$$\theta_x = \frac{\phi_x d_x}{\phi_b d_b + \sum_{y \in X} \phi_y d_y} \tag{8}$$

and

$$\theta_b + \sum_{x \in X} \theta_x = 1 \tag{9}$$

Then the probability of observing $n_4$ counts at mass 204, $n_6$ counts at mass 206 and $n_b$ counts of background is given by

$$p(n_4, n_6, n_b | \theta_4, \theta_6, \theta_b) = \frac{(n_4 + n_6 + n_b)!}{n_4! n_6! n_b!} \theta_4^{n_4} \theta_6^{n_6} \theta_b^{n_b} \tag{10}$$

Whereas the observations $n_4$, $n_6$ and $n_b$ are integers, the parameters $\theta_4$, $\theta_6$ and $\theta_b$ are decimal numbers that are constrained to a constant sum. In other words, they belong to the simplex. Thus, we can map the three multinomial parameters to two logratio parameters, thereby establishing a natural link between counting data and compositional data. For example:

$$\beta_6^4 \equiv \ln\left(\frac{\phi_4 - \phi_b}{\phi_6 - \phi_b}\right) = \ln\left(\frac{\theta_4/d_4 - \theta_b/d_b}{\theta_6/d_6 - \theta_b/d_b}\right) \text{ and} \tag{11}$$

$$\beta_6^b \equiv \ln\left(\frac{\phi_b}{\phi_6 - \phi_b}\right) = \ln\left(\frac{\theta_b/d_b}{\theta_6/d_6 - \theta_b/d_b}\right) \tag{12}$$

$\beta_6^4$ and $\beta_6^b$ can be estimated from $n_4$, $n_6$ and $n_b$ by the method of maximum likelihood. See the Appendix for further details.

The normal and logistic normal distributions are controlled by two sets of parameters: location parameters and shape parameters. In the case of the normal distribution, the location parameter is the mean and the shape parameter is the standard deviation (or covariance matrix). In contrast, the multinomial distribution has only one set of ($\theta$) parameters. The precision of multinomial counts is governed by the number of observed counts ($\sigma[n] = \sqrt{n}$). More sophisticated models are possible when the observed dispersion of the data exceeds that which is expected from the multinomial counting statistics (e.g., Galbraith and Laslett, 1993; Vermeesch, 2018b). However in this paper we will assume that such *overdispersion* is absent from the standards, and from the single-spot analyses.





## 9 Dead-time correction

It takes a few tens of nanoseconds for a secondary electron multiplier to record the arrival of an ion. During this 'dead-time', the detector is unable to register the arrival of additional ions. This phenomenon can significantly bias isotope ratio estimates that include high intensity ion beams. Fortunately, the dead-time effect can be easily corrected. It suffices that the dwell times are adjusted by the cumulative amount of time that the detectors were incapacitated. Let $d'_x$ be the 'effective dwell time' of ion beam $x$:

$$d'_x = d_x - n_x d_x \tag{13}$$

where $d_x$ is the dead time of the detector that measures $x$. Then the expected normalised beam counts can be redefined as:

$$\theta'_x = \frac{\phi_x d'_x}{\phi_b d'_b + \sum_{y \in X} \phi_y d'_y} \tag{14}$$

and the normalised beam intensities as

$$\phi'_x = \frac{\theta_x / d'_x}{\theta_b / d'_b + \sum_{y \in X} \theta'_y / d_y} \tag{15}$$

## 10 Within-spot drift correction

Thus far we have assumed that all ions are measured synchronously, which is the case in multicollector instruments. However in single collector instruments, the measurements are made asynchronously. This can cause biased results if the signals drift over time. In SHRIMP data processing, it is customary to correct this drift by normalising to a secondary beam monitor signal (Bodorkos et al., 2020). A unified data reduction algorithm for SHRIMP and Cameca instruments requires a different approach, in which the time dependency of the signals is parameterised using a log-linear model. For Faraday detectors:

$$n_x^i = \text{bkg} + \mathcal{N}(\exp[\alpha_x + \gamma_x \tau_x^i], \sigma^2) \tag{16}$$

where $n_x^i$ is the ion beam intensity of the $i^{\text{th}}$ integration for mass $x$ evaluated at time $\tau_x^i$, $\sigma$ is the standard deviation of the normally distributed Johnson noise (which is to be estimated from the scatter of the data around the best fit line), and 'bkg' is the background signal. This is usually a nominal value for Cameca instruments and an actual set of measurements ($n_b^i$) for SHRIMP data. For SEM detectors, the scatter of the data around the log-linear fit is controlled by Poissonian shot noise:

$$n_x^i \sim \text{bkg} + \text{Pois}\left(\exp[\alpha_x + \gamma_x \tau_x^i]\right) \tag{17}$$

The blank-corrected signal ratio of two ion beams $x$ and $y$ (evaluated at $\tau_x^i$) can then be drift corrected as follows:

$$^i\beta_x^y \equiv \ln\left(\frac{\phi_y^i - \phi_b^i}{\phi_x^i - \phi_b^i}\right) + \gamma_y\left(\tau_x^i - \tau_y^i\right) \tag{18}$$

where $\phi_x^i$ and $\phi_y^i$ are the dead time corrected normalised beam intensities for the $i^{\text{th}}$ integration of masses $x$ and $y$, respectively, and $\phi_b^i$ is the corresponding blank value. See the Appendix for further details.



## 11 Fractionation

Mass spectrometer signals are recorded in volt (for Faraday detectors) or Hertz (or secondary electron multipliers). The age
equation, however, requires atomic ratios. In general, signal ratios do not equal atomic ratios, because they are affected by two
types of fractionation:

1. Mass-dependent fractionation: The Pb-isotopes span a range of four mass units, with $^{208}$Pb being 2% heavier than
   $^{204}$Pb. Both the production and detection efficiency of secondary ions varies with atomic mass, and significant errors can
   potentially occur if the resulting mass fractionation is uncorrected for. Mass fractionation can be quantified by comparing
   the measured signal ratios of a reference material with its known isotopic ratio. This is easy to do in a log-ratio context
(Vermeesch, 2015).

2. Elemental fractionation: The fractionation between the Pb-isotopes is caused by (slight) differences in their physical
   properties, i.e. their mass. As briefly mentioned in Section 4, much stronger fractionation effects tend to occur between
   the isotopes and Pb and U, because they are not only physically, but also chemically different. These chemical differences
   affect the complex processes that occur when the primary ion beam interacts with the target material (Williams, 1998).

In the context of SIMS U–Pb geochronology, mass-dependent fractionation is commonly ignored, because the most impor-
tant isochemical ratio is that between $^{206}$Pb and $^{207}$Pb, which lie within 0.5% mass units of each other. This is unresolvable
given typical analytical uncertainties. The mass fractionation is greater for $^{204}$Pb, but so it its analytical uncertainty. Therefore,
the atomic $^{204}$Pb/$^{206}$Pb, $^{207}$Pb/$^{206}$Pb and $^{208}$Pb/$^{206}$Pb ratios can be directly estimated from the (drift corrected) $^{204}$Pb/$^{206}$Pb,
$^{207}$Pb/$^{206}$Pb and $^{208}$Pb/$^{206}$Pb signal ratios. This is not the case for the $^{206}$Pb/$^{238}$U and $^{208}$Pb/$^{232}$Th-ratios, which are affected by
255 strong elemental fractionation effects. This fractionation expresses itself in two ways.

1. Within-spot fractionation

   Over the course of a SIMS spot analysis, the Pb/U and Pb/Th ratio changes as a function of time. This elemental
   fractionation can be modelled using a log-linear model that is similar to that used for the within-spot drift correction:

   $$^{i}\beta_x^y = {}^{0}\beta_x^y + \gamma_x^y \tau_x^i \tag{19}$$

where $^{0}\beta_x^y$ is the inferred logratio of the blank-corrected signals at 'time zero', which can be found using the method of
   maximum likelihood (see Appendix). With Equation 19, the isotopic logratios can be interpolated (or extrapolated) to
   any point in time ($\tau$):

   $$^{\tau}\beta_x^y = {}^{0}\beta_x^y + \gamma_x^y \tau \tag{20}$$

   The most precise values of $^{\tau}\beta_x^y$ are obtained when $\tau$ is chosen in the middle of the analytical sequence. These values can
be used for subsequent calculations. Alternatively, we can also use the time-zero intercepts $^{0}\beta_x^y$.





2. Between-spot fractionation

The Pb/U and Pb/Th signal ratios may vary between adjacent spots on the same isotopically homogenous reference material. This fractionation obeys the power law relationship given by Equation 2. Expressing this formula in terms of corrected signal ratios:

$$\ln\left[\frac{(\phi_6' - \phi_b') - (\phi_4' - \phi_b')(6/4)_c}{\phi_u' - \phi_b'}\right] = A + B\ln\left[\frac{\phi_o' - \phi_b'}{\phi_u' - \phi_b'}\right] \tag{21}$$

where $(6/4)_c$ stands for the $^{206}$Pb/$^{204}$Pb-ratio of the common Pb (see the footnote of Section 4). Recasting Equation 21 in terms of the interpolated logratio estimates:

$$\ln\left[\exp(^\tau\beta_u^6) - \exp(^\tau\beta_u^4)(6/4)_c\right] = A + B\,^\tau\beta_u^o \tag{22}$$

where '$o$' stands for the uranium oxide ($^{238}$U$^{16}$O$_2^+$ or $^{238}$U$^{16}$O$^+$) and '$u$' stands for $^{238}$U$^+$.

## 12 U–Pb age calculation

Having applied Equation 22 to a reference material with known atomic $^{206}$Pb/$^{238}$U-ratio $\left[^{206}\text{Pb}/^{238}\text{U}\right]_a^{st}$, the atomic $^{206}$Pb/$^{238}$U-ratio of the sample is given by

$$\ln\left[\frac{^{206}\text{Pb}}{^{238}\text{U}}\right]_a^{sm} = \ln\left[\frac{^{206}\text{Pb}}{^{238}\text{U}}\right]_a^{st} + {}^\tau\beta_u^6(sm) - A - B\,^\tau\beta_u^o(sm) \tag{23}$$

where $^\tau\beta_u^6(sm)$ and $^\tau\beta_u^o(sm)$ are the interpolated logratio estimates of the sample. The $^{206}$Pb/$^{238}$U-age is then obtained by plugging $\left[^{206}\text{Pb}/^{238}\text{U}\right]_a^{sm}$ into the age equation. Uncertainties are obtained by standard error propagation (see Appendix).

## 13 Examples

The following sections will illustrate the SIMS U–Pb data reduction process using two datasets:

1. Dataset 1 was acquired by Dr. Yang Li at IGG-CAS Beijing, using a Cameca 1280HR instrument. It uses Temora zircon (Temora2, 416.8±1.1 Ma, Black et al., 2004) as a reference standard and 91500 zircon (1062.4±0.2 Ma, Wiedenbeck et al., 1995) as a sample. Measurements consist of seven sweeps through a set of 11 mass-stations per single spot measurement, for $^{90}$Zr$_2$O (0.48 second dwell time), $^{92}$Zr$_2$O (0.08 s), mass 200.5 (background, 4.00 s), $^{94}$Zr$_2$O (0.32 s), $^{204}$Pb (4.96 s), $^{206}$Pb (2.96 s), $^{207}$Pb (6.00 s), $^{208}$Pb (2.00 s), $^{238}$U (2.96 s), ThO$_2$ (2.96 s), and UO$_2$ (2.96 s).

2. Dataset 2 was acquired by Dr. Simon Bodorkos at Geoscience Australia using a SHRIMP-II instrument. It also uses Temora zircon as a reference standard, and 91500 zircon and OG.1 (3440.7±3.2 Ma, Stern et al., 2009) and M127 (524.36±0.16 Ma, Nasdala et al., 2016) as samples. Measurements consist of six sweeps through a set of 10 mass-stations per single spot measurement, for $^{90}$Zr$_2$O (2.0 second dwell time), $^{204}$Pb (20 s), mass 204.04091 (background, 20 s), $^{206}$Pb (15 s), $^{207}$Pb (40 s), $^{208}$Pb (5 s), $^{238}$U (5 s), ThO (2 s), UO (2 s), and UO$_2$ (2 s).







**Figure 3.** Time resolved signals (counts) of (a) Temora zircon (spot `Tem@6`) analysed by a Cameca 1280HR instrument at IGG-CAS Beijing; (b) M127 zircon (spot `M127.1.2`) analysed by a SHRIMP-II instrument at Geoscience Australia.

Figure 3 shows the time resolved SEM counts of one representative spot measurement for each dataset. Side-by-side comparison of these two datasets reveals some interesting similarities and differences. All of the high intensity signals exhibit clear

transient behaviour, which is caused by changes in oxygen availability that occur during primary ion bombardment (Magee et al., 2017). The transience of the individual SEM signals biases the isotopic ratios. For example, there are 109 seconds between the $^{204}$Pb and $^{208}$Pb measurements in each SHRIMP cycle. During these 109 seconds, the $^{208}$Pb signal drops on average by 2%, resulting in an equivalent bias of the $^{204}$Pb/$^{208}$Pb ratio (Section 10). A nearly identical drop per cycle is observed for the Cameca data.

However, there is a key difference between the Cameca and SHRIMP datasets. For the Cameca data, the within-spot signal drift of the U- and Pb-isotopes is the same, with both having a negative slope in the example of Figure 3. But for the SHRIMP data, the U- and Pb-drifts act in opposite directions: the $^{207}$U-signal exhibits an increase in sensitivity with time, whereas the Pb-signal decreases in intensity with time. This marked difference in behaviour between the two instruments reflects a





difference in their design, causing a difference in the energy window of the secondary ions analysed (Ireland and Williams,
2003).

For Faraday detectors, the within-spot drift correction uses a generalised linear model with a lognormal link function (Equation 16). For SEM data, it uses a Poisson link function (Equation 17). In either case, the model enforces strictly positive isotopic abundances. Isotopes of the same element (such as $^{204|6|7|8}$Pb) have the same slope parameter, but different intercepts. Isotopes of different elements are free to have different slopes (Figure 4).

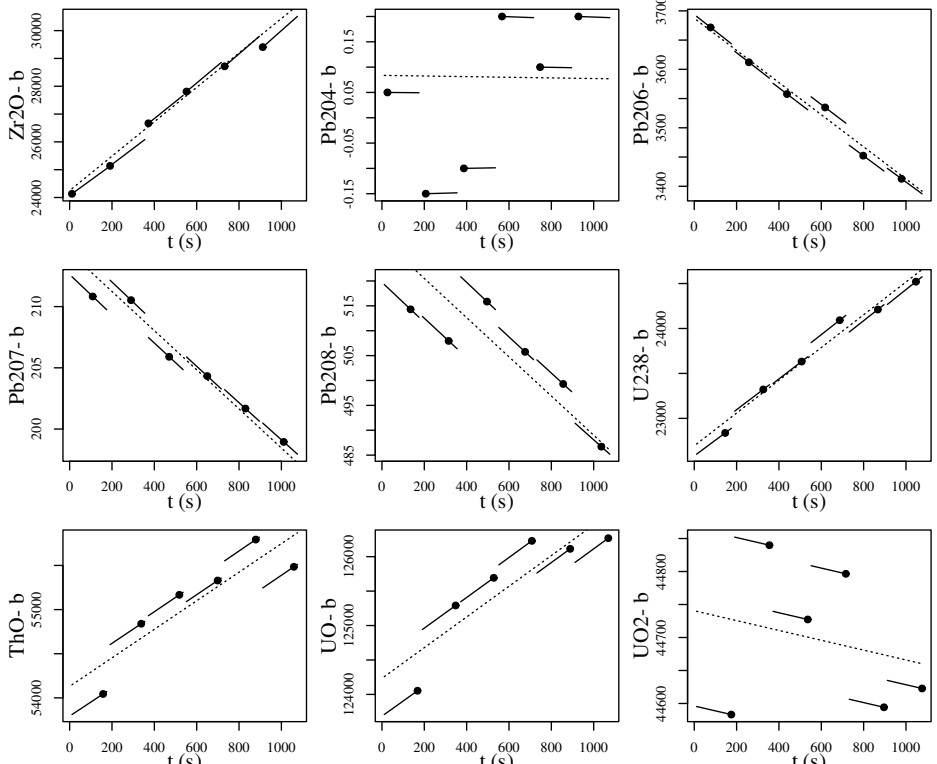

**Figure 4.** Within-spot drift correction of blank-corrected SHRIMP data. Dotted lines are log-linear functions (Equation 17) whose slopes are used for the drift correction but for no other purpose. Solid lines mark the duration of each mass spectrometer sweep, with the black dots representing the starting point of each individual mass station within the sweeps. The solid lines are parallel to the dotted lines (in log space) and show how the asynchronous mass spectrometer signals can be translated in time to extract synchronous isotopic ratios. Vertical axes have units of counts per second.

Figure 5 applies another log-linear function (Equation 19) to model the within-spot fractionation of Temora spot 11. This function models the drift-corrected logratios as a linear function of analysis time. The slope of the log-linear functions are a





function of the elemental fractionation between the numerator and denominator elements. Because there is no fractionation between two isotopes of the same element, the slope of the Pb/Pb ratios is zero.

The ability of logratio statistics to avoid negative ratios is apparent from the first panel of Figure 5. Even though some of

the ratios of the blank-corrected signals are zero or negative (because the blank exceeded the signal), the generalised linear fit is strictly positive. The natural ability of compositional data analysis to rule out negative ratios avoids many problems further down the data processing chain.

The right hand side of Figure 5 maps the four (log)ratios back to five equivalent raw signals (one for each isotope). The last two panels of the figure show how the logratio approach manages to effectively capture subtle fluctuations of the U and UO

signal intensities.

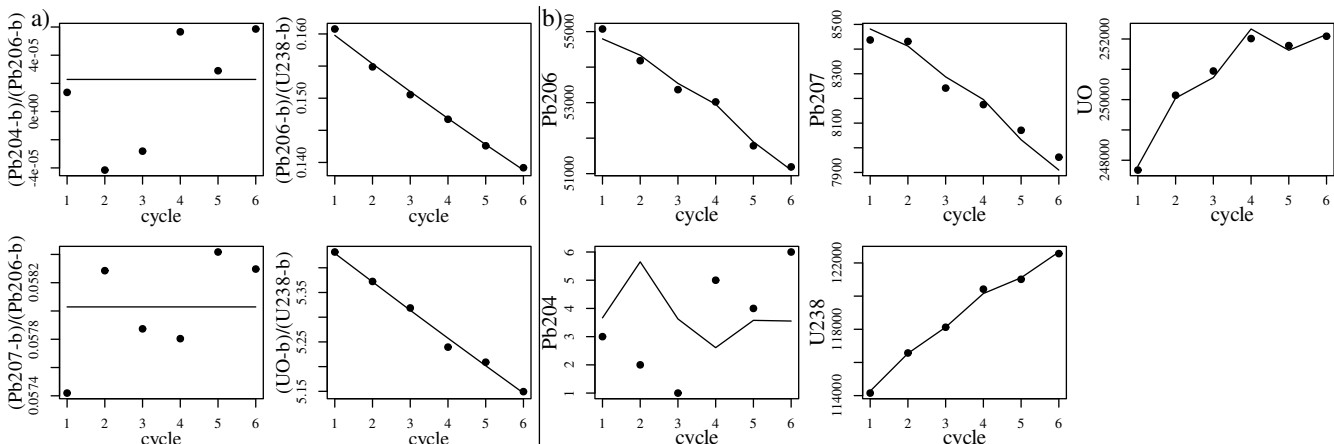

**Figure 5.** a) blank- and drift-corrected ratio fits of the SHRIMP data, obtained using the generalised linear model of Equation 19. Vertical axes are unitless. b) the four (log)ratio fits can easily be converted back to five isotope signal fits using the inverse logratio transformation. Vertical axes have units of counts.

The logratio intercepts obtained by Equation 19 form a linear array of calibration data (Equation 22). Figure 6.a fits a straight line through these points using the linear regression algorithm of York et al. (2004). Alternatively, instead of fitting a calibration line through the logratio intercepts ($\tau = 0$, Figure 6.a), it is also possible to interpolate or extrapolate the logratio composition to any other point in time. For example, the green ellipses in Figure 6.b show the inferred logratio compositions at 544 seconds

(i.e., $\tau = 544$), which represents the midpoint of the analyses. The slope of this calibration line is notably different than that obtained by fitting a line through the compositions at 0 seconds. This change in slope reflects the different mechanisms that are responsible for elemental fractionation within and between SIMS spots.

Figure 7 applies Equation 23 to 91500 zircon, using the Temora data for calibration. It shows only the purely random errors, i.e. ignoring the uncertainty of the standard calibration. Including the calibration errors does not only inflate the uncertainties,

but also causes inter-sample error correlations (Figure 1). To demonstrate this phenomenon, let us revisit the Cameca data,





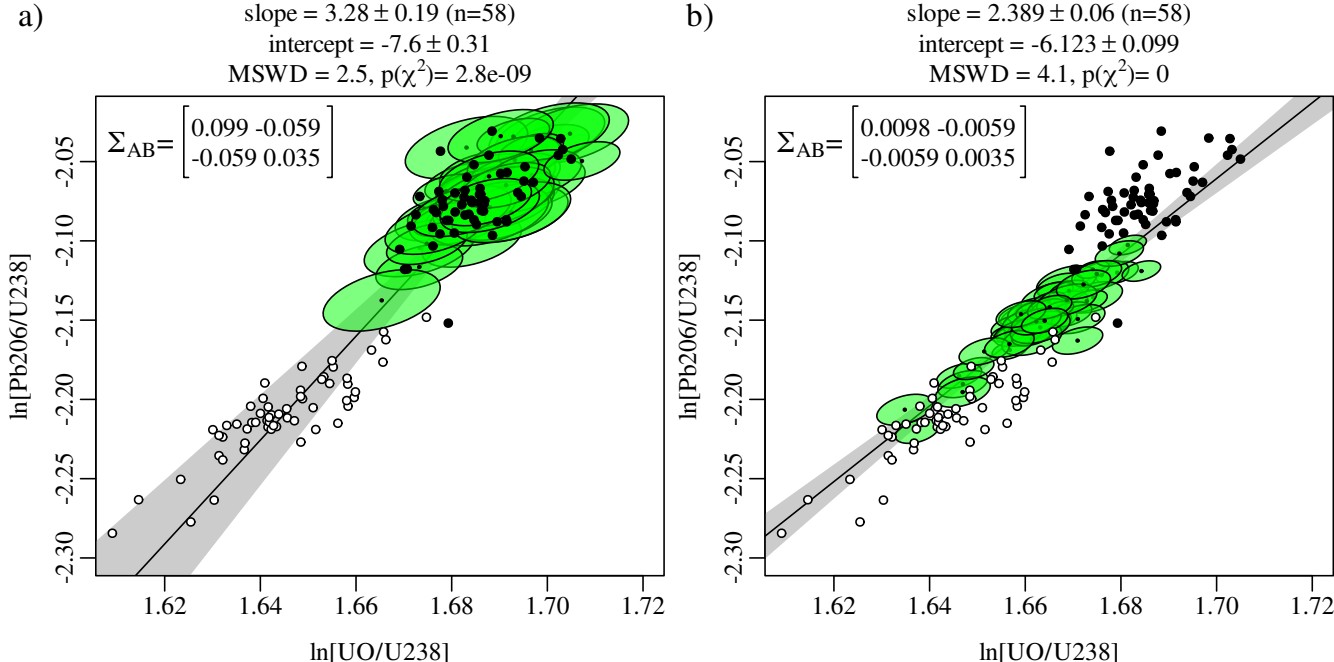

**Figure 6.** a) U–Pb calibration curve for the Temora SHRIMP data using the time-zero ($\tau = 0$) intercepts (green ellipses). Black and white dots mark the first and last sweeps of each analysis, respectively. b) the logratio data and calibration fit for the same data, but evaluated at the midpoint ($\tau = 544$), resulting in a more precise calibration. $\Sigma_{AB}$ marks the covariance matrix of the intercept $A$ and slope $B$. These matrices show that the analytical uncertainties of the intercept and slope are nearly perfectly correlated with each other ($r[A, B] \approx -1$).

using 91500 for the calibration curve, and Temora as a sample (Figure 8). Table 2 shows the uncertainty budget of four selected aliquots from this sample.

|   | int. unc. (%) | tot. unc. (%) | $r[*, b]$ | $r[*, c]$ | $r[*, d]$ |
|---|---|---|---|---|---|
| $a$ | 0.32 | 0.70 | 0.62 | 0.36 | -0.34 |
| $b$ | 0.31 | 0.44 |  | 0.36 | -0.17 |
| $c$ | 0.30 | 0.36 |  |  | 0.019 |
| $d$ | 0.45 | 0.56 |  |  |  |

**Table 2.** Uncertainty budget of the four Temora zircon analyses highlighted in Figure 8. The first data column shows the standard errors of the calibrated $^{206}$Pb/$^{238}$U-ratios ignoring the uncertainty of the calibration fit (i.e., using internal uncertainties only). The second column shows the total error including the external uncertainty associated with the calibration fit. The upper triangular matrix shown in the remaining three columns contain the (total) error correlations of the four aliquots.





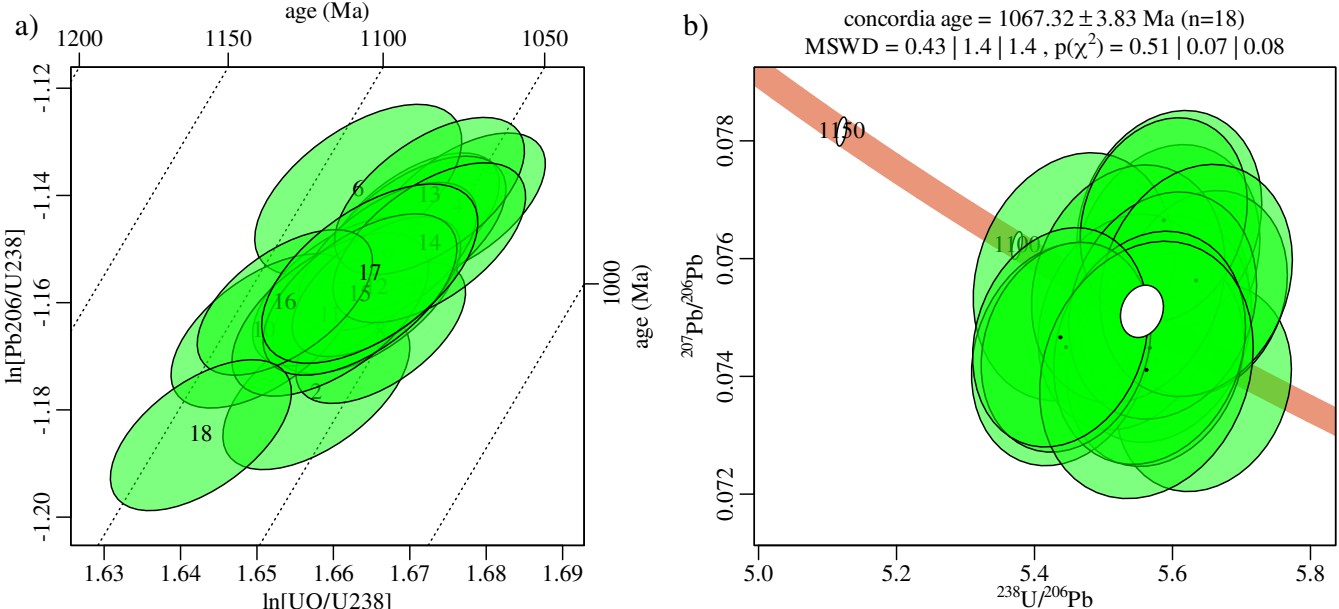

**Figure 7.** a) Calibration plot of the SHRIMP 91500 zircon data. Dotted lines are parallel to the best fit line of Figure 6b. b) The same data shown on a Tera-Wasserburg concordia diagram, which was obtained with `IsoplotR` and does not take into account systematic uncertainties associated with the calibration fit. All uncertainties are shown at 95% confidence. MSWD and p-values represent the goodness of fit for equivalence, concordance, and equivalence + concordance.

Propagating the systematic uncertainties increases the error estimates (Table 2) by different amounts for different spots. For aliquot $c$, which is located immediately below the mean of the 91500 data, the calibration uncertainty only mildly increases the standard error from 0.30% to 0.36%. However, for aliquot $a$, which is horizontally offset from the mean of the calibration data, the systematic calibration uncertainty more than doubles the standard error from 0.32% to 0.70%. The calibration error also causes the standard errors of the various aliquots to be correlated with each other. For example, the total uncertainties of aliquots $a$ and $b$ are positively correlated ($r[a, b] = 0.62$) because their UO$_2$/U-ratios are both offset from the mean of the calibration in the same direction. In contrast, the uncertainties of aliquots $a$ and $d$ are negatively correlated ($r[a, d] = -0.34$) because they are offset in opposite directions from the mean of the calibration data.

The inter-sample error correlations are important when calculating weighted means (Vermeesch, 2015) and isochrons. Taking into account the full covariance structure of the data benefits both the accuracy and the precision of any derived age information. To take full advantage of this statistical power will require the development of a new generation of high level data reduction software. For example, future versions of `IsoplotR` (Vermeesch, 2018a) will accept full covariance matrices as input. A comprehensive discussion of this topic falls outside the scope of this paper.





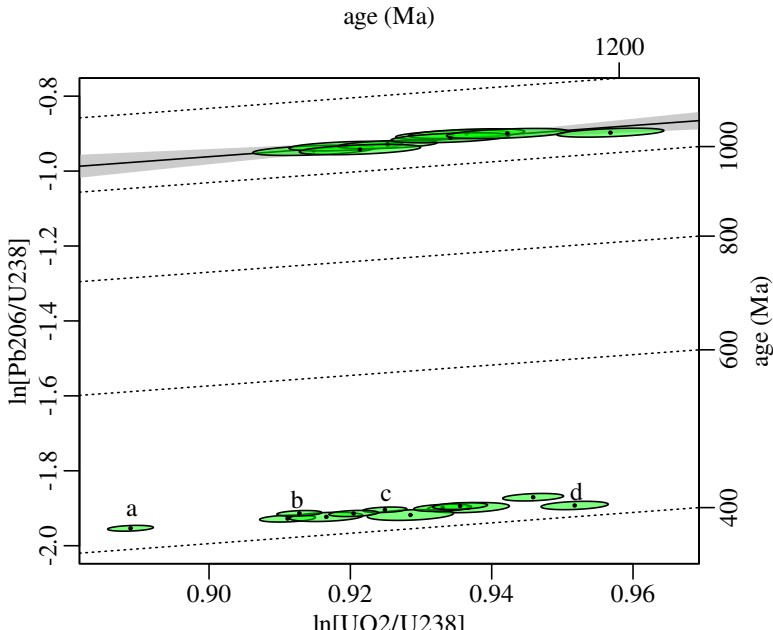

**Figure 8.** Calibration curve of the Cameca data, using 91500 zircon as a standard (top half of the plot), and Temora zircon as a sample (bottom half). *a–d* mark four Temora aliquots whose uncertainty budget is explored in Table 2.

## 14 The `R` package `simplex`

`simplex` is an `R` package for SIMS data processing that implements the algorithm presented in this paper. The program can be run in three modes: online, offline, and from the command line. The online version can be accessed at http://isoplotr.es.ucl. ac.uk/simplex/. It contains three example U–Pb datasets, including the two datasets used in this paper, plus a Cameca monazite

U–Th–Pb dataset.

`simplex` currently accepts raw data as Cameca `.asc` and SHRIMP `.op` and `.pd` files. Support for SHRIMP `.xml` files will be added later. The online version is a good place to try the look and feel of the software. However, it is probably not the most practical way to process lots of large data files. For a more responsive user experience, `simplex` can also be run natively on any operating system (Windows, Mac OS or Linux). To this end, the user needs to install `R` on their system (see

https://r-project.org/ for details). Within `R`, the `simplex` package can be installed from the `Github` code-sharing platform using the `remotes` package, by entering the following commands at the console:

```
install.packages("remotes")
remotes::install_github("pvermees/simplex")
```

Once installed, `simplex`' graphical user interface (GUI) can be started by entering the following command at the console:





```
simplex::simplex()
```

A third and final way to use `simplex` is from the command line. This allows advanced users to create automation scripts and extend the functionality of the package. `simplex` comes with an extensive API (Application Programming Interface) of fully documented user functions. An overview of all these functions can be obtained by typing the following command at the console:

```
help(package="simplex")
```

## 15   Discussion

This paper introduced a new algorithm for SIMS U–Pb geochronology, in which raw mass spectrometer signals are processed using a combination of logratio analysis and Poissonian counting statistics. In contrast with existing data reduction protocols, the new algorithm simultaneously processes all the aliquots in an analytical sequence. It thereby produces an internally consistent set of isotopic ratios and their associated covariance matrix. This covariance matrix is a rich source of information that captures both random and systematic uncertainties, including inter-sample error correlations that have hitherto been ignored in geochronology.

The example data of Section 13 showed that these inter-sample error correlation can be either positive or negative (see also Vermeesch, 2015; McLean et al., 2016). Ignoring them degrades both the accuracy and precision of high end data processing steps such as isochron regression and concordia age calculation. Unfortunately, existing postprocessing software such as `Isoplot` (Ludwig, 2003) and `IsoplotR` (Vermeesch, 2018a) does not yet handle inter-sample error correlations. Further work is needed to extend these codes and take full advantage of the new algorithm. `IsoplotR` was designed with such future upgrades in mind: its input window contains a large number of spare columns that will accommodate covariance matrices in a future update. Once the aforementioned high end data reduction calculations have been updated, it will be possible to fully quantify the gain in precision and accuracy of the new algorithm compared to the previous generation of SIMS data reduction software.

The data reduction principles laid out in this paper are applicable not only to U–Pb geochronology, but also to other SIMS applications such as stable isotope analysis. In fact, `simplex` already handles such data for multicollector Cameca instruments. It is worth mentioning that the stable isotope functionality can also be used to correct $^{207}$Pb/$^{206}$Pb ratio measurements for mass-dependent isotope fractionation, as was briefly discussed in Section 11. Future updates of the mass-dependent fractionation correction will also addresses the overcounted blank problem that was mentioned in Section 6.

Besides U–Pb geochronology and stable isotopes, the new data reduction paradigm can also be adapted to other chronometers and other mass spectrometer designs, such as Thermal Ionisation Mass Spectrometry (TIMS, Connelly et al., 2021), noble gas mass spectrometry (Vermeesch, 2015), and LAICPMS (McLean et al., 2016). `simplex` already includes a function to export data to `IsoplotR`. Adding similar functionality to other data processing software will improve geochronologists' ability to



integrate multiple datasets whilst keeping track of systematic uncertainties, including those associated with age standards and decay constants.

*Code and data availability.* The source code, installation instructions and example datasets for `simplex` can be accessed at https://github.com/pvermees/simplex/

## Appendix A: Appendix

This Section provides further algorithmic details for the new U–Pb data processing workflow. It assumes that ions are recorded on SEM detectors, which is by far the most common configuration. The case of Faraday collectors is similar and, in fact, simpler.

Within-spot drift is modelled using a log-linear function (Equation 17) with a distinct intercept ($\alpha_x$) for each ion channel ($x$), and shared slopes ($\gamma_X$) between isotopes of the same element ($X$). To illustrate this concept, consider the case of $^{204|6|7}$Pb (inclusion of $^{208}$Pb is a trivial extension). Let $\hat{n}_x^i$ be the time dependent parameter of the shot noise for $^{20x}$Pb, where $x \in \{4, 6, 7\}$:

$$\hat{n}_x^i = \mathrm{bkg} + \exp(\alpha_x + \gamma_{Pb} t_x^i) d'_x \tag{A1}$$

then the log-likelihood function for the parameters is given by:

$$\mathcal{LL}_d \left( \alpha_{\{x\}}, \gamma_{Pb} \right) = \mathrm{Const.} + \sum_x \sum_{i=1}^N \left( n_x^i \ln \left[ \hat{n}_x^i \right] - \hat{n}_x^i \right) \tag{A2}$$

where $N$ represents the number of sweeps. The parameters $\alpha_{\{x\}} = \{\alpha_4, \alpha_6, \alpha_7\}$ and $\gamma_{Pb}$ are estimated by maximising $\mathcal{LL}_d$ with respect to them. Only $\gamma_{Pb}$ is used in subsequent calculations. The intercepts $\alpha_{\{x\}}$ are discarded.

The next step of the data reduction extracts logratios from the raw data using a log-linear model that is similar to the within-spot drift correction (Equation 19). Here, in contrast with the drift correction, the intercepts are just as important as the slopes. For isochemical ratios such as $^{207}$Pb/$^{206}$Pb, the slope of the drift-corrected logratios is zero and we only need to estimate the intercept. For multichemical ratios such as $^{238}$U/$^{206}$Pb, both the slope and the intercept are non-zero. In order to keep track of covariances, it is useful to process all the isotopes together, using a common denominator. For example, using $^{206}$Pb ('6') as a common denominator and $^{204}$Pb ('4'), $^{207}$Pb ('7'), $^{238}$U ('$u$') and UO ('$o$') as numerators:

$$^i\beta_6^x = {}^0\beta_6^x + \gamma_6^x \tau_6^i + \gamma_X \left( \tau_x^i - \tau_6^i \right) + \ln[d'_6] - \ln[d'_x] \tag{A3}$$

where '$X$' stands for Pb if $x \in \{4, 6, 7\}$, for U if $x = u$, and for UO if $x = o$. Then the normalised ion counts are given by:

$$\theta_y^i = \exp \left[ {}^i\beta_6^y \right] / D_i \quad \text{and} \quad \theta_6^i = 1/D_i \tag{A4}$$





for $y \in \{4, 7, u, o\}$, with

$$D_i = 1 + \sum_y \exp[{}^i\beta_6^y] + \frac{n_b^i}{\sum_z n_z^i} \tag{A5}$$

in which $z \in \{4, 6, 7, u, o, b\}$. Then the log-likelihood is calculated as:

$$\mathcal{LL}_l\left({}^0\beta_6^{\{x\}}, \gamma_6^{\{x\}}\right) = \text{Const.} + \sum_{i=1}^{N} \sum_x n_x^i \ln[\theta_x^i] \tag{A6}$$

where $\gamma_6^4 = \gamma_6^7 = 0$ because there is no elemental fractionation between the Pb-isotopes. From the logratios with common denominator, it is easy to derive any other logratio:

$${}^\tau\beta_x^y = {}^\tau\beta_6^y - {}^\tau\beta_6^x \tag{A7}$$

One of the main advantages of the new data reduction method is its ability to keep track of the full covariance structure of
the data, including inter-sample error correlations. This ability is derived from the fact that all parameters are derived by the method of maximum likelihood, which stipulates that the approximate covariance matrix of any set of estimated parameters can be obtained by inverting the negative matrix of second derivatives (i.e., the Hessian matrix) of the log-likelihood function with respect to said parameters:

$$\Sigma \approx -\mathcal{H}^{-1} \tag{A8}$$

For example, to estimate the covariance matrix of the logratio slopes and intercepts for a single spot analysis, the Hessian is a $6 \times 6$ matrix that includes the second derivatives of $\mathcal{LL}_l$ with respect to $\beta_6^4$, $\beta_6^7$, $\beta_6^u$, $\beta_6^o$, $\gamma_6^u$ and $\gamma_6^o$. Computing this matrix is tedious to do by hand but straightforward to do numerically.

Given the covariance matrix of the logratios, subsequent data reduction steps propagate the analytical uncertainties by conventional first order Taylor approximation. Thus, if $y = f(x)$, then:

$$\Sigma_y \approx J_f \Sigma_x J_f^T \tag{A9}$$

where $J_f$ is the Jacobian matrix (and $J_f^T$ its transpose) of partial derivatives of $f$ with respect to $x$. For example, to estimate the $m \times m$ covariance matrix of $m$ fractionation-corrected ${}^{206}\text{Pb}/{}^{238}\text{U}$-ratios, error propagation of Equation 23 would involve an $m \times (2m+3)$ Jacobian matrix and an $(2m+3) \times (2m+3)$ covariance matrix containing the uncertainties of $A$, $B$, $\ln\left[\frac{{}^{206}\text{Pb}}{{}^{238}\text{U}}\right]_a^{st}$, as well as ${}^\tau\beta_u^6(j)$ and ${}^\tau\beta_u^o(j)$ (for $j$ from 1 to $m$).

*Author contributions.* PV is the sole author of this paper

*Competing interests.* The author is an Associate Editor of *Geochronology*



*Acknowledgements.* The idea for this work was born during an academic visit of the author to the Institute of Geology and Geophysics (IGG) at the Chinese Academy of Sciences (CAS) in Beijing. With no prior experience in handling SIMS data, the author relied on the expertise of Dr. Yang Li and Dr. Li-Guang Wu (IGG-CAS) to introduce him to the world of Cameca data processing, and of Dr. Simon

Bodorkos, Dr. Charles Magee and Dr. Andrew Cross (Geoscience Australia) for SHRIMP data processing. The test data were kindly shared by Dr. Li and Dr. Bodorkos, who also provided detailed feedback on the manuscript prior to submission. Software development for the `simplex` data reduction program was supported by NERC Standard Grant #NE/T001518/1 ('Beyond `Isoplot`'), which aims to develop an 'ecosystem' of inter-connected geochronological data processing software. The graphical user interface for `simplex` makes extensive use of the `shinylight` and `dataentrygrid` packages developed by software engineer Tim Band, who is employed on this NERC

grant.





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
