# Peer review of "An algorithm for U–Pb geochronology by secondary ion mass spectrometry"

_Geochronology, 2022_

## Author Response (AR1)

Dear Dr. McLean,

I hereby submit the revised version of my GChron manuscript on "An algorithm for U-Pb geochronology by secondary ion mass spectrometry". I have already addressed the reviewers' major comments in my response on the GChron discussion page. In this cover letter, I will go over their specific comments.

Pieter Vermeesch

Morgan Williams

*"L9: typo "handes" should be "handles"."*

Fixed.

*"L19: The series of short sentences here are succinct but do not read well; rephrasing along the lines of 'In contrast to SIMS instrumentation, LAICPMS instruments are built by numerous manufacturers, and the widely used data reduction software packages are generally compatible with all of their data formats"."*

The short sentences have been replaced with: "LAICPMS instrumentation is built by numerous manufacturers, whose data files are compatible with a rich ecosystem of data reduction codes such as Iolite, Glitter and LADR. This facilitates the intercomparison of different laboratories, different instrument designs and so forth. In contrast, the SIMS U–Pb world is dominated by just two manufacturers, whose SHRIMP (Sensitive High Resolution Ion Micro-Probe) and Cameca instruments use completely separate data reduction protocols."

*"L122, L126: Specifically refer to the additive log ratio (ALR) used."*

Done.

*"L142: The term 'trick' here perhaps confounds the intention and doesn't lend to confidence in the use of it. When referring to generalization, perhaps refer to Eqn. 4 (which is what is generalized, as the logratio approach is not specific to a dimensionality, even if examples are 2/3 dimensional)."*

This has been replaced with "The ALR transformation of Equation 4 can easily be generalised to more than three components. For example, if 207 Pb is added to the mix, then the four-component 204|6|7 Pb–238 U-system can be mapped to the three component ln(204|6|7 Pb/238 U)-space (Figure 2)."

*"Figure 2a – I think the labels on the upper figure may be reversed. This might also be better termed a 'projection' from a four-component simplex (which would be a tetrahedron). This diagram depicts the compositional data aspect of the manuscript and software rather nicely."*

Figure 2 has been redrafted from scratch.

*"Eqn 13, Section 9: Dead-time correction – I think this exclusively applies to a non-extended deadtime correction? Generally, the difference between extending/non-extending is likely to be small except for particularly high count-rates (maybe for every high U, or analysing a minor element on an EM), but it might be worth noting this, at the author's discretion."*

This is correct. The text has been revised as follows: "where dx is the (non extended) dead-time of the detector that measures x."

*"L224: SHRIMP data processing – Depending on the scenario, I think Dodson interpolation may also be used to similar effect for dealing with smooth intensity changes during cycles of data collection within a single spot (Dodson, 1978)."*

This has been replaced with "In SHRIMP data processing, it is customary to correct this drift by normalising to a secondary beam monitor (SBM) signal[2] , followed by double interpolation of numerator and denominator isotopes (Bodorkos et al., 2020; Dodson, 1978)." with a footnote that reads: "SBM normalisation is not a magic bullet. For example, whereas the 206Pb signal of SHRIMP spot M127.1.2 rises with time (Figure 3b), its corresponding SBM signal drops."

*"Figure 5: Depending on the desired manuscript formatting, the capitalization of this figure caption may be off (e.g. "a) Blank …")?"*

Corrected.

*"L321: "Figure 6.a" – the references to figures may be easier to read without the period (e.g. Figure 6a; as is used in the Figure 7 caption)."*

Done.

*"Fig7b: It may be worth noting that the white ellipse represents the concordia age and uncertainty."*

The caption now says: "The white ellipse marks the weighted mean composition, with MSWD and p-values representing the goodness of fit for equivalence, concordance, and equivalence + concordance."

All other comments have been addressed in the online discussion.

Nicole Rayner

*"Line 75 […] I suggest the last sentence of this paragraph be removed."*

Done.

*"Line 86 […] Suggest use of "reference material" throughout"*

Done. I have also changed the abbreviation "st" to "r" (and "sm" to "s") in the equations.

*"Caption Figure 1 – "calibration error" should be calibration uncertainty"*

Done.

*"108 [...] I suggest you refer to ['sample'] as "analytical spot"."*

Done, throughout the revised manuscript.

*"Line 129 consider illustrating Table 1 data in a ternary diagram prior to mapping to Euclidean space (data points on Figure 2 perhaps?)."*

I understand the reviewer's suggestion, but decided not to follow it. With 8 figures, the paper is already quite heavy on graphics, and I feel that a 9th figure would be too much. I would also prefer not to clutter the important Figure 2 with additional information. I did, however, redraft it from scratch to fix some inaccuracies.

*"Line 150 – here "inter-sample" should be "inter-spot""*

Inter-sample has been replaced by inter-spot throughout the revised manuscript, except for one instance (in section 4), which does refer to multiple samples.

*"Line 164 – typo "black" instead of "blank""*

Typo fixed. Thank you!

*"Section 7 "Zeros" – consider merging with the blanks section [...] Since section 7 is so brief, I am not even very clear why it is needed."*

-> Section 7 was created in symmetry with the enumerated list of issues in the introduction. Since it is clear that this symmetry escaped the attention of the reviewer, I have followed her advice and merged Sections 6 and 7.

*"Sections 8 and 9 – as a SIMS mass spectrometrist/geochronologist and not a statistician I am searching for points of familiarity, which up to this point I am largely able to do. You lose me here in these sections. More direct explanation of the steps in traditional data reduction that are replaced by this approach and then how these values get used/incorporated into ratios would be helpful. Part of the difficulty in following is that in the traditional data reduction approach the deadtime correction happens first, but in the paper "Deadtime" follows the section about "Dealing with count data" which seems counterintuitive."*

The "Deadtime" section follows the "Dealing with count data" section because it builds on it. Equations 14 and 15 are redefining Equations 7 and 8 of the previous section. The reviewer's confusion stems from a fundamental difference between the new method and conventional data reduction algorithms, which is explained in a new paragraph:

"There is a fundamental difference between this approach and existing SIMS data reduction approaches. Conventional data reduction algorithms apply the dead-time correction to the raw data, before calculating isotopic data. In the new method, the dead-time correction is applied to $\theta$ and $\varphi$ , which are unknown parameters that must be estimated from the data."

*"Line 250 states that mass-dependent fractionation is commonly ignored. It has been established from long term reproducibility studies that this is not true, and some labs do not ignore it"*

The paper says that "mass-dependent fractionation is commonly ignored". It does not claim that it is ALWAYS ignored. To clarify this point, I have added a reference to Stern et. al. (2009), who present one of the few counter-examples where mass-dependent fractionation is accounted for:

"In the context of SIMS U–Pb geochronology, mass-dependent fractionation is commonly ignored (but not always, e.g., Stern et al., 2009), because the most important isochemical ratio is that between 206Pb and 207Pb, which lie within 0.5% mass units of each other."

*"Line 283, refer to it as Temora2 throughout the manuscript"*

Done.

*"Line 293 refer to mass spec "cycles", elsewhere "sweeps". I prefer cycle"*

-> All instances of "sweep" have been replaced with "cycle".

*"Line 301, not negative, positive"*

-> Well spotted. Thanks.

*"Line 302 "207" superscript in error"*

-> Corrected.

*"Line 302 Is this treatment instead of SBM norm or before/after? I think in lieu but I don't understand how this might be affected by drift in the primary beam intensity which might either enhance or minimize the depth/oxygen availability effect illustrated in this diagram. For example if over the time of the analysis the primary beam intensity decreases then increases, the within-spot drift may be U-shaped, not linear and thus a single slope regression applied to each cycle isn't appropriate."*

I investigated the SBM method in more detail and discussed the issue with SHRIMP expert Dr. Simon Bodorkos. It turns out that SBM normalisation was really important for early SHRIMP models, but is less important for modern instruments. It is not a magic bullet either. For example, whereas the 206 Pb signal of SHRIMP spot M127.1.2 rises with time (Figure 3b), the corresponding SBM signal drops. This point is raised in a new footnote to Section 9.

*"Line 307 – the strict enforcement of a positive value for 204-b doesn't reflect the real-life behaviour of ion-counted data and could indicate a real problem with the analytical setup. I am concerned that this approach "hides" a real problem."*

This is a valid concern. The revised version of simplex marks problematic aliquots (for which the blank corrected logratio drifted to negative infinity) by asterisk. It is now also possible to fix overcorrected 204 signals by overriding the nominal blank value.

*"Line 312 – insert "within-spot" when referring to fractionation here for clarity."*

Done.

*"Caption 5 – since side a is blank and drift corrected (eg. Pb204 – b/Pb206 - b), are the converted counts shown in part b also blank corrected (in which case the vertical axis should be labelled Pb204 – b, Pb206 – b etc)"*

Actually, no: the fitted signals in panel b include the blank.

*"Line 328 – edit text to "Figure 7 applies the Pb/U calibration (Equation 23) to 91500…." Makes it easier for the reader to know "Equation 23" [what] does without having to go back to check."*

Good suggestion. Done.

*"Line 330 – again "inter-sample" when should be inter-spot (reminder to rationalize this usage throughout)"*

Inter-sample has been replaced with inter-spot throughout the text except for one instance which did effectively relate to samples instead of aliquots/spots.

*"Line 331 – figure 7 uses Temora2 as the reference material and 91500 as the sample. Figure 8 uses 91500 as the reference material and Temora2 as the sample. Both of these materials are commonly used as RM's and so switching their usage back and forth is tricky for the reader. Perhaps there is one but I don't see any reason why Figure 8 can't be recast with Temora2 as the RM which would streamline things."*

I understand the comment. However, the reason why the sample and reference material are swapped in Figure 8 is that the reference material exhibits greater spread than the sample. This amplifies the error correlations that this figure aims to illustrate. There are two ways to follow the reviewer's suggestion:

1) use 91500 as a reference material in Figure 6 and Temora zircon as a sample in Figure 7: this would cause Figure 6 to be a cluttered mess of overlapping error ellipses.

2) use Temora as a reference material and 91500 as a sample in Figure 8: this would reduce the error correlations in Table 2, thereby reducing the impact of the point that this table hopes to make.

Neither of these solutions is good, so I think that it is better to keep things as they are. However, to avoid any confusion, I have added some clarification to the text:

"let us revisit the Cameca data and swap the sample and reference materials around"

*"Figure 7 caption – specify "using Temora2 as the reference material" at the end of the first sentence."*

Done. This will also alleviate the issue raised in the reviewer's previous comment.

*"Line 368 – I'm not sure what is meant by "In contrast with existing data reduction protocols, the new algorithm simultaneously processes all the aliquots in an analytical sequence." Please clarify."*

This has been rephrased as follows:

"In contrast with existing data reduction protocols, which handle each aliquot of an analytical sequence separately, the new algorithm simultaneously processes all of them in parallel."

*"Line 381 – While I appreciate that not all the benefits of the covariance matrices can currently be accomplished, it would be great to see a comparison of results between the previous approaches and this one one using the provided datasets."*

The following example has been added to the revised manuscript:

"For example, the conventional weighted mean can be replaced with a matrix expression that accounts for correlated uncertainties (Equation 92 of Vermeesch, 2015). Applying this modified algorithm to the positively correlated aliquots a and b of Table 2 changes the weighted mean from −2.6922 to −2.6854 and increases the standard error of that mean from 0.37% to 0.44%. Applying the same calculation to the negatively correlated aliquots a and d changes their weighted mean from −2.7083 to −2.7076 and reduces its uncertainty from 0.43% to 0.35%."

Response to the Associate Editor report

You correctly observed that my writing style was modelled after that of Ken Ludwig. I understand that this is not everybody's cup of tea. However, I personally found Ludwig's paper really clear and, for the most part, easy to translate into computer code. Similarly, I have found the work of Rex Galbraith tremendously useful. His papers are even more concise than those of Ludwig, and use a mixture of Greek and Latin symbols. Once you get used to this notation, I find it easier to follow than a more complex set of 'meaningful' but more verbose symbols.

I know that you have experimented with a different writing style in your excellent "Algorithms and software for U-Pb geochronology by LA-ICPMS". However, as you pointed out, this more accessible writing style has not increased the impact of the latter paper. In my experience, only a disappointingly small proportion of the people who cite my papers actually read them. And those people tend to be the numerate ones who are not afraid of some mathematical notation.

I did however address your other recommendations:

I followed your suggestions for Lines 8, 9, 10, 17

Equation 1 includes Pb204 because this turns the U-Pb age equation into a ternary system, thereby setting the stage for Section 5 and Figure 2. In order to make the link with ternary systems, I have changed the introductory sentence to Section 2 to: "The common Pb corrected 206 Pb/ 238 U-age (t) depends on the relative abundances of three nuclides, 204Pb, 206Pb and 238U"

Table 2: thanks for the excellent observation. I replaced mA with pA.

Lines 57 and 58: I reformulated these as suggested.

*Line 67 -- "Include a brief explanation of the magnitude of difference you expect with a logratio approach"*

The relative difference between the geometric and arithmetic mean is given by 1-sqrt(exp(v)) where v is the variance of the logratios. However, in order not to further overload the paper with equations, I have addressed your comment by rephrasing the penultimate sentence of Section 2: "Inaccurate 206Pb/238U-ratios inevitably result in inaccurate U–Pb dates, with the degree of inaccuracy scaling with the relative standard deviation of the measured data."

Line 73 now provides an example for the Student-t multiplier (-2.262 for a=0.025 and df=9).

*Line 75 -- "In conversation with those in the SIMS U-Pb field, including many of its pioneers, I have heard the following argument: negative isotope ratios/abundances are ok in the presence of noise and censoring or forcing them to be positive will bias averages derived from the noisy data."*

It is awkward to mention these personal communications in a peer-reviewed paper. The fact that none of these private opinions have been formally written down probably reflects the fact that their proponents know that they are built on shaky foundations. In any case, I disagree with the notion that negative ratios are ok, and that it is necessary to allow them to reduce bias. It is possible that biases arise when symmetric residuals are truncated at zero. The logratio transformation fixes this issue by allowing skewed residuals. However, in a different context, it is true that negative ratios cannot be ignored. So I have added the following sentence to Section 3: "Negative isotope ratios can also arise from blank and baseline corrections, but this will be further discussed in Section 6."

*Line 91 -- The statement around "both random and systematic" is too simplistic. Shared systematic uncertainties like the U decay constants do not need to be propagated to compare SIMS U-Pb and TIMS U-Pb, but the TIMS tracer uncertainty and SIMS primary standard age uncertainty (sans decay constants and perhaps sans tracer calibration) will need to be propagated.*

This is a good point. This paragraph has been reformulated as follows:
"Great care must be taken which sources of uncertainty should or should not be included in the error propagation. In some cases, inter-sample comparisons of SIMS U–Pb data may legitimately ignore systematic uncertainties. However, when comparing data acquired by different methods, both random and some systematic uncertainties must be accounted for. For example, when comparing U–Pb and 40 Ar/39 Ar data, the decay constant uncertainty must be propagated; and when comparing SIMS and TIMS U–Pb data, SIMS primary standard age uncertainty and the TIMS tracer uncertainty must be propagated. The conventional way to tackle inter-sample comparisons is called 'hierarchical' error propagation (Renne et al., 1998; Min et al., 2000; Horstwood et al., 2016). Under this paradigm, the random uncertainties are processed first, and the systematic uncertainties afterwards."

*Equation 2 -- Is this 206Pb/238U ratio corrected for Pbc?*

It depends. If the reference material has a uniform (uncorrected) 206Pb/238U ratio, and if this ratio is known, then the uncorrected ratio can be used. If neither of these conditions are fulfilled (e.g. because only the age of the reference material has been reported in the literature, and not the actual U-Pb composition), then a common-Pb correction is needed. Both of these options are available in simplex. This complexity is briefly mentioned in footnote 1. Does this not suffice as an explanation?

*Also, the 2 in parentheses for the species 238U16O(2) is confusing -- I'd leave discussion of different calibration schemes to later in the article.*

Omitting the (2) would upset the Cameca people. I have clarified in the meaning of the subscript in the revised text: "where the subscript m stands for the measured signal ratio, which is generally different from the atomic ratio; and the subscript (2) refers to the fact that the calibration normally uses uranium oxide for SHRIMP and uranium dioxide for Cameca instruments."

*Line 145 -- Change "into very" to "into the very"*

Done.

*Line 148 -- This reference to "the full covariance matrix" is a bit vague.*

I have followed your suggestion and added the following sentence to this paragraph: "For example, four components yield three logratios, which require a 3 × 3 covariance matrix to describe the uncertainties and uncertainty correlations."

*Line 150 -- Note that you'll need an expanded logratio covariance matrix to capture the inter-spot error correlations, whose size scales with the number of spots that will be interpreted together.*

I have added the following two sentences to bring this point to the attention of the reader: "... In that case, the covariance matrix must be expanded to accommodate multiple spots (e.g., Vermeesch, 2015). For example, the covariance structure of three spots in a four component system can be captured in a 9 × 9 matrix."

*Figure 2 -- Include the necessary info needed to recreate this diagram in the caption -- the 238/235, 206/204, and 207/204 ratios, as well as the assumption that they are the same for all age regions. Also, please use the figure caption to explain the "regional geography" of these plots. What do the boundaries of the colored regions mean? Where is the familiar concordia curve (on the right)? [...] A more helpful caption might include a discussion of strengths/weaknesses of each plot (e.g., mixing lines are straight on left, elliptical confidence regions for log-normally distributed uncertainties on right). Finally, if the purpose of this plot is to illustrate the compositional nature of U-Pb geochronology data and its transformation to a log-ratio space where calculations are made, then this should be articulated in the caption.*

The caption was extended as follows: "The U–Pb age equation (a) projected onto the four-component simplex, and (b) mapped to a three-dimensional Euclidean logratio space. $^{235}U$ is omitted from the diagrams because it exists in a constant ratio to $^{238}U$ ($^{238}U/^{235}U=137.818$, Hiess et al., 2012). The boundaries between the coloured regions mark mixing lines between radiogenic and inherited end-member compositions, assuming common Pb ratios of $[^{207}Pb/^{204}Pb]_c = 10$ and $[^{206}Pb/^{204}Pb]_c=9$. The mixing lines define isochron surfaces that are linear in the simplex and curved in logratio space. Rotating the logratio plot 90 degrees clockwise produces a logarithmic version of the familiar Wetherill concordia diagram. Concordant $^{206}Pb/^{238}U$ and $^{207}Pb/^{235}U$ compositions are marked by a thick black line from 0 to 4Ga, and by a dotted line beyond 4Ga. This paper makes the case that U-Pb data processing is best done in logratio space, because this 'liberates' the data from the geometric constraints of the simplex, producing symmetric uncertainty distributions and more accurate results."

*To make this figure even more informative, add a third ternary field to the left-hand plot with 206Pb-238U-207Pb, so that it is more analogous to the back face of the plot on the right and so that a concordia line will be shown on a ternary plot.*

I haven't added a third panel, but I did replace the 207Pb-204Pb-238U panel with the 206Pb-238U-207Pb diagram. I also added the concordia line as a thick black line (from 0 to 4Ga) and a dotted line (>4Ga), as mentioned in the revised caption:

[Figure]

*Also, the right-hand plot has its back panel coordinate axes reversed from the (Wetherill) concordia convention that has 206/238 on the y-axis and 207/235 on the x-axis.*

This is true, but rotating the figure 90 degrees clockwise produces a plot that looks uglier (in my opinion), as shows one project above the back panel rather than under it:

[Figure]

*An even better idea might be to plot 207/206 vs. 238/206 in log-ratios on the right, as the Tera-Wasserburg concordia plot is more widely used in the SIMS U-Pb community.*

I have also tried this, but the problem here is that the 207/206 vs. 238/204 plot contains lines that cross over and cannot easily be colour coded. It would, of course, be possible to omit the 207/206 vs. 238/204 plot, but then it is no longer possible to show the logratio plot as a 3D image:

[Figure]

*Line 170 -- Change "can" to "can be"*

Done.

*Line 186 -- Add a line to describe what "normalized" means in this context*

But this was already described by the equation provided immediately below.

*Line 212 -- The bias happens for ratios between high- and low-intensity ion beams. The dead time effects on two high-intensity ion beams with the same average intensity will cancel out.*

Very true. I rephrased this as follows: "This phenomenon can significantly bias isotope ratio estimates that include ion beams of contrasting intensity."

*Equation 13 -- I believe the first dx on the RHS of this equation is a dwell time and the second is the dead-time*

Thanks for spotting this! The dead-time was supposed to be rendered as a Greek delta.

*Equation 13 -- This expression is the same as the conventional deadtime equation for a measured intensity, corr = meas/(1-dt\*meas) where corr and meas are corrected and measured intensities and dt is the deadtime. Their equivalence was not obvious to me until I did some calculations on my own. This I think is at odds with the assertion on line 221 that "there is a fundamental difference between this approach and existing SIMS data reduction approaches." This is certainly true for the the log/Poisson link functions and drift corrections, but would need further support here.*

I think that you are right for stationary beams but not for ion beams that undergo log-linear drift. In any case, the sentence about the 'fundamental difference' between the new approach and the conventional approach was added in response to Reviewer 2. I have removed the words "fundamentally different" from the new version.

*Line 225 -- Multicollector machines can and do run single-collector routines.*

This has been rephrased as "Thus far we have assumed that all ions are measured synchronously, which is the case in multicollector configurations. However in single collector experiments, the measurements are made asynchronously."

*Equation 16 -- For clarity, describe what the background is. The manuscript also refers frequently to the 'blank', as defined (?) on line 38. The equations use 'b' and 'bkg'. If (as I understand) these are the same, please harmonize their use in text and equations. For Faraday detectors, the bkg as written would also include the reference voltage for that Faraday, which might be below zero.*

I used 'blank' because of my previous experience with noble gas mass spectrometry. I have replaced the word 'blank' with 'background' throughout the text.

*Equations 16 and 17 -- Identifying alpha and gamma in a figure (Figure 4) and referencing that figure would be very helpful.*

The intercepts alpha are not used, but the slopes gamma are now mentioned in the caption of Figure 4.

*In Figure 4, it took me a long time to understand (?) that the gamma and alpha of equation 17 correspond to the dotted lines only, and not to the solid lines -- perhaps switch which is solid and dotted?*

According to the caption: "Dotted lines are log-linear functions whose element-specific slopes ($\gamma_X$ in Equation 17) are used for the drift correction but for no other purpose. Solid lines mark the duration of each mass spectrometer cycle, with the black dots representing the starting

point of each individual mass station within the cycles. The solid lines are parallel to the dotted lines (in log space) and show how the asynchronous mass spectrometer signals can be translated in time to extract synchronous isotopic ratios."

*It is also unclear to me, based on the (conflicting?) explanations here and in the Appendix (lines 406-414) whether you use species- (e.g., ion channel-) specific or element-specific gammas in the within-spot drift correction.*

Thanks for pointing out this inconsistency in the notation. I have replaced $\gamma_x$ with $\gamma_X$ in the main text, which is now using the same notation as the Appendix. The revised text also clarifies that: "$n^i_x$ is the ion beam intensity of the ith integration for mass $x$ of element $X$ [...] Note that the intercept parameters ($\alpha_x$) are ion-specific, whereas the slopes ($\gamma_X$) are element-specific. See the Appendix for an example."

*Line 233 -- There will be Johnson and shot noise on the Faraday detector.*

This line was reformulated as follows: "$\sigma$ is the standard deviation of the normally distributed data scatter around the best fit line"

*Equation 19 -- I suggest the multiply-indexed parameters be explained/illustrated in a concise figure, with or without accompanying data.*

All my attempts to explain the procedures graphically produced figures that are more complicated than the equations. Here is one attempt to capture Equations 18 (panel a) and 19/20 (panel b). I think that it will confuse rather than enlighten the reader:

[Figure]

I did add the following clarification about the relationship between the equations of Sections 9 and 10: "Note that $\gamma_X^Y=0$ if $X=Y$ and, if the sensitivity drift of the data is smooth and closely follows the trends defined by Equations 96 and 17, then $^0\beta_x^y \approx \alpha_y - \alpha_x$ and $\gamma^Y_X \approx \gamma_Y - \gamma_X$."

*Line 287 -- Please provide more details about the Cameca 1280HR -- is it a single collector? Is that collector an ion counter? Likewise on line 292 for the SHRIMP II. Please specify all times listed are for a single "mass station", not the total time during the analysis.*

I have added the following clarification to the text: "a Cameca 1280HR instrument with five SEMs and two Faraday detectors that was used in single collector mode, using an SEM for all mass-stations. [...] 0.48 second dwell time per cycle [...] using a single collector SHRIMP-II instrument, employing an SEM for all mass-stations.

*Line 305 -- Clarify if by "the same" you mean they have the same slopes are the slopes have the same sign.*

This has been rephrased as "For the Cameca data, the U and Pb signals both drift in the same direction, resulting in positive slopes for the example of Figure 4"

*Figure 4 -- Please specify which analysis from the SHRIMP dataset this figure shows.*

They are the same data as in Figure 3 (which is now called Figure 4). This has been clarified in the revised caption.

*See comments on equations 16 -- 17. Why does this figure have time on the x-axis and Figure 5 have 'cycle' on the x-axis?*

This is how simplex does it, because it makes it easier to remove outliers. I have replaced the cycle numbers with the actual timestamps of the different mass stations in the revised manuscript.

*Figure 5 -- Can you explain here or near Equation 19 how the parameterization you use results in a piecewise continuous line?*

The origin of the piecewise continuous line is explained in the new caption: "Multiplying the normalised ion beam intensity fits with the total number of counts per sweep allows direct comparison with the raw measurements, which are shown as filled circles."

*Why do the ratios have a discontinuous derivative when the intensities (in my understanding) are fit with the single dotted (?) line in Figure 4?*

The dotted line in Figure 4 (now Figure 5) is only used for the drift correction, and not for the logratio fit. This is explained in the new Figure 3.

*Is there a way to illustrate uncertainties in this figure?*

This would require that I visualise the Poisson errors as asymmetric error bars. I believe that this would do more harm than good because most geochronologists equate error bars with standard errors obtained from replicate measurements. The counting errors are a different type of

uncertainty. I have added the following sentence to the caption: "Note that the measurements have Poisson uncertainties, which scale with the square root of the values."

*Line 336 -- Specify the specific systematic uncertainties that increase error estimates by different amounts.*

The following clarification was added to the text: "Propagating the calibration curve uncertainty increases the error estimates"

*Figure 6 -- Please specify whether slope, intercept, and data ellipse uncertainties are ±1 or 2 sigma (I think the first two are 1, the latter 2). I don't understand the point made in the figure caption about the correlation of the slope and intercept uncertainties.*

All the uncertainties are now reported at 2 sigma, as mentioned in the revised caption. I have removed the covariance matrix of the slope and intercept.

*Equation A2 -- I'm unfamiliar with the LL notation. Is this just log-likelihood? If so, define in line 411.*

I don't understand this question. LL is defined in line 411 (now 425).

---

## Author Response (AR2)

**Prof. Pieter Vermeesch**
University College London
+44 (0)20 3108 6369
http://ucl.ac.uk/~ucfbpve/

2 August 2022

Dear Prof. McLean,

I have modified line 91 and Figure 2 of my revised manuscript according to your instructions. Thanks again for the excellent editorial handling of this paper.

Sincerely yours,

Pieter Vermeesch